# Outcomes Associated with Respiratory Failure for Patients with Cardiogenic Shock and Acute Myocardial Infarction: A Substudy of the CULPRIT-SHOCK Trial

**DOI:** 10.3390/jcm9030860

**Published:** 2020-03-20

**Authors:** Maria Rubini Giménez, P. Elliott Miller, Carlos L. Alviar, Sean van Diepen, Christopher B. Granger, Gilles Montalescot, Stephan Windecker, Lars Maier, Pranas Serpytis, Rokas Serpytis, Keith G. Oldroyd, Marko Noc, Georg Fuernau, Kurt Huber, Marcus Sandri, Suzanne de Waha-Thiele, Steffen Schneider, Taoufik Ouarrak, Uwe Zeymer, Steffen Desch, Holger Thiele

**Affiliations:** 1Department of Internal Medicine/Cardiology, Heart Center Leipzig at University of Leipzig and Leipzig Heart Institute, 04289 Leipzig, Germany; maria.rubini@usb.ch (M.R.G.); marcus.sandri@medizin.uni-leipzig.de (M.S.); Steffen.Desch@medizin.uni-leipzig.de (S.D.); 2Cardiovascular Research Institute Basel (CRIB), University Hospital Basel, 4031 Basel, Switzerland; 3Division of Cardiovascular Medicine, Yale University School of Medicine, New Haven, CT 06520-8017, USA; Elliott.miller@yale.edu; 4Yale National Clinician Scholars Program, New Haven, CT 06510-8088, USA; 5The Leon H. Charney Division of Cardiology, New York University School of Medicine, New York, NY 10016, USA; carlos.alviar@nyulangone.org; 6Department of Critical Care Medicine and Division of Cardiology, Department of Medicine, University of Alberta, Edmonton, AB 8440, Canada; sv9@ualberta.ca; 7Cardiology Department, Duke Clinical Research Institute, Durham, NC 27701, USA; christopher.granger@duke.edu; 8Department of Cardiology, Institut de Cardiologie (AP-HP), Hôpital Pitié Salpêtrière, Sorbonne Université, ACTION study group, 75013 Paris, France; gilles.montalescot@psl.aphp.fr; 9Department of Cardiology, Inselspital, Bern University Hospital, University of Bern, 3010 Bern, Switzerland; stephan.windecker@insel.ch; 10Department of Cardiology, University Hospital Regensburg, 93077 Regensburg, Germany; lars.maier@ukr.de; 11Faculty of Medicine, Vilnius University and Vilnius University Hospital Santaros Klinikos, 08410 Vilnius, Lithuania; Pranas.Serpytis@santa.lt (P.S.); rokas.serpytis@sana.lt (R.S.); 12Department of Cardiology, Golden Jubilee National Hospital, Glasgow G81 4DY, UK; keith.oldroyd@nhs.net; 13Cardiology Department, University Medical Center Ljubljana, 1000 Ljubljana, Slovenia; marko.noc@mf.uni-lj.si; 14Cardiology Department, University Heart Center Luebeck, 23538 Luebeck, Germany; georg.fuernau@uksh.de (G.F.); Suzanne.deWaha@uksh.de (S.d.W.-T.); 153rd Department of Internal Medicine, Cardiology and Intensive Care Medicine, Wilhelminenhospital and Sigmund Freud University, Medical School, 2301 Vienna, Austria; kurt.huber@meduniwien.ac.at; 16Statistical department, Institut für Herzinfarktforschung, 67063 Ludwigshafen, Germany; schneider@stiftung-ihf.de (S.S.); ouarrak@stiftung-ihf.de (T.O.); 17Hospital of the city of Ludwigshafen, Medical Clinic B and Institut für Herzinfarktforschung, 67063 Ludwigshafen, Germany; uwe.zeymer@t-online.de

**Keywords:** cardiogenic shock, respiratory failure, mechanical ventilation, non-invasive ventilation

## Abstract

Background: Little is known about clinical outcomes of patients with acute myocardial infraction (AMI) complicated by cardiogenic shock (CS) requiring mechanical ventilation (MV). The aim of this study was to identify the characteristics, risk factors, and outcomes associated with the provision of MV in this specific high-risk population. Methods: Patients with CS complicating AMI and multivessel coronary artery disease from the CULPRIT-SHOCK trial were included. We explored 30 days of clinical outcomes in patients not requiring MV, those with MV on admission, and those in whom MV was initiated within the first day after admission. Results: Among 683 randomized patients included in the analysis, 17.4% received no MV, 59.7% were ventilated at admission and 22.8% received MV within or after the first day after admission. Patients requiring MV had a different risk-profile. Factors independently associated with the provision of MV on admission included higher body weight, resuscitation within 24 h before admission, elevated heart rate and evidence of triple vessel disease. Conclusions: Requiring MV in patients with CS complicating AMI is common and independently associated with mortality after adjusting for covariates. Patients with delayed MV initiation appear to be at higher risk of adverse outcomes. Further research is necessary to identify the optimal timing of MV in this high-risk population.

## 1. Background

Patients with cardiogenic shock (CS) due to acute myocardial infarction (AMI) experience respiratory failure requiring mechanical ventilation (MV) in 80% of the cases [1,2,3,4] being a marker of higher patient complexity [2,5]. Current clinical management of respiratory failure in CS is largely based on expert opinion, preclinical data or small clinical series, and most commonly includes invasive mechanical ventilation (IMV) [6,7]. Alternatives such as non-invasive ventilation (NIV) are frequently used in acute pulmonary edema in patients without shock, but its use and effects in CS are less well established [6,8]. 

Although frequently lifesaving, implementation of mechanical ventilation (MV) requires proper understanding of cardiopulmonary interactions between respiratory mechanics and hemodynamics, particularly in patients with CS, as well as familiarity and knowledge about the potential beneficial and adverse effects [1]. As such, the effects of MV in improving hemodynamics by unloading the left ventricle, improving oxygenation and tissue perfusion, may play a beneficial role in CS, particularly when initiated early in the clinical course. However, despite the significant interactions with the cardiovascular system and pervasiveness in CS, the ideal management of respiratory failure, understanding of the risk factors, and the development of associated clinical outcomes or complications have to date not been adequately investigated [9,10]. Given the paucity of available data, we sought to explore the association between MV timing and strategy among patients with CS from the CULPRIT-SHOCK (Culprit Lesion Only PCI versus Multivessel PCI in Cardiogenic Shock) randomized clinical trial.

## 2. Methods

### 2.1. Trial Design

The present report is a secondary analysis of the multicenter randomized CULPRIT-SHOCK trial (www.clinicaltrials.gov NCT01927549). The design details including inclusion and exclusion criteria have been published previously [11,12,13]. Briefly, patients with AMI complicated by CS and multivessel coronary artery disease with planned early revascularization by percutaneous coronary intervention (PCI) were randomly assigned 1:1 to undergo either culprit-lesion-only PCI (with possible staged revascularization) or immediate multivessel PCI. CS was defined by a systolic blood pressure of less than 90 mmHg for more than 30 min or the use of vasopressor agents to maintain a systolic pressure of at least 90 mmHg, along with clinical signs of pulmonary congestion, and signs of impaired organ perfusion (altered mental status, cold and clammy skin and limbs, oliguria or arterial lactate level over 2.0 mmol/L). Patients were excluded if they were older than 90 years of age or if the etiology of CS included a mechanical cause, massive pulmonary embolism or single vessel coronary disease. In addition, patients were excluded if they had undergone cardiopulmonary resuscitation for over 30 min with no intrinsic heart action and poor neurological status, if they had known severe renal insufficiency, if their life expectancy was thought to be less than 6 months from concomitant severe disease, and if the onset of shock was over 12 h before randomization. Respiratory support utilization, including type of ventilator support (NIV vs. IMV) and timing of MV was left at the discretion of the treating clinician.

All subjects gave their informed consent for inclusion before they participated in the study. The study was conducted in accordance with the Declaration of Helsinki, and the protocol was approved by the local Ethics Committee. Ethical approvals by the lead ethical committees for each country are: (a) Germany, Ethical Committee at the University of Luebeck: reference number 13-142; (b) Netherlands, Medisch Ethische Toetsingscommissie (Academisch Medisch Centrum, University of Amsterdam): reference number E2-170; (c) Austria, Magistratsabteilung 15-Gesundheitsdienst der Stadt Wien: reference number EK-13-241-0214; (d) Lithuania, Lietuvos Bioetikos Komitetas: reference numbers L-14-01/1 and L-14-01/2; (e) France, Comité de Protection des Personnes, Ile de France 1: reference number 2014-janvier.-13464; (f) Poland, Klinika Intensywnej Terapii Kardiologicznej: reference number IK-NP-0021-97/1408/13; (g) Slovenia, Komisija Republike Slovenije za medicinsko etiko: reference numbers 63/12/13 and 60/09/14; (h) Switzerland, Kantonale Ethikkommission Bern (KEK): reference number 041/14; (i) Italy, Comitato Etico Provinciale di Reggio Emilia: reference number 2013/0029992; (j) Belgium, Universiteit Antwerpen (Ethics Committee): reference number 15/11/116; (k) UK, National Health Service (NHS) (Scotland Research Ethics Committee): reference number 14/YH/0116; and (l) Scotland, NHS (Scotland rEsearch Ethics Committee): reference number 14/SS/0072.

### 2.2. Outcomes

The primary endpoint of the current subanalysis was death from any cause within 30 days after randomization. Patients were stratified in three groups: (1) no MV (not requiring MV at all), (2) MV present on admission and (3) MV initiated within the first day after admission (or the following days). Secondary endpoints of the current subanalysis included renal failure requiring renal replacement within 30 days, re-hospitalization for congestive heart failure, repeat revascularization at 30 days, time to hemodynamic stabilization, use and duration of catecholamine therapy, intensive care unit (ICU) length of stay and duration of MV, respectively. 

Safety endpoints included stroke and bleeding, which were defined as bleeding type 2, 3 or 5 on the Bleeding Academic Research Consortium scale [13,14]. All endpoints were adjudicated by blinded clinical events committee members.

### 2.3. Statistical Analysis

Categorical variables were expressed as counts and percentages and were compared by Chi^2^-test. Continuous variables were presented as median and interquartile range (IQR) and were compared by Mann-Whitney-Wilcoxon test. The primary endpoint of death from any cause within 30 days after randomization was displayed by Kaplan-Meier-curves. Adjusted odd ratios (OR) for the primary endpoint in the ventilated groups (compared to no-ventilated groups) were also calculated (adjusted for age > 73 years, male, weight (kg), resuscitation within 24 h before randomization, left bundle branch block, ST-segment elevation, ST-segment depression, heart rate (bpm), creatinine on admission (umol/l) and triple vessel disease). To identify predictors of MV within 30 days, logistic regression models were constructed. The multivariate models included all baseline variables with a relevant association (*p*-value < 0.1) from univariate analyses.

In addition, we performed two sensitivity analyses. First, we assessed for the association of NIV and clinical outcome by stratifying our cohort into four groups: (1) no respiratory support, (2) NIV alone, (3) NIV before IMV, and (4) IMV alone. Second, given the ubiquitous use of MV in patients who experience cardiac arrest, we assessed for the association between mortality and MV use in a cohort that excluded patients presenting with cardiac arrest.

All *p*-values were two-tailed and *p* < 0.05 was considered statistically significant unless stated otherwise. All analyses were performed using SAS statistical package version 9.4 (Cary, NC, USA).

## 3. Results

### 3.1. Patient Characteristics

Among the 683 patients included from the CULPRIT-SHOCK trial, 408 (59.7%) presented with MV at admission, 156 (22.8%) received MV within the first day or later after admission due to respiratory failure, and 119 (17.4%) did not require any MV (Figure 1). Patients undergoing MV were younger, had a higher body mass indices (BMI), higher heart rate, presented more often with clinical signs of impaired organ perfusion, had more frequently undergone resuscitation within 24 h, had worse renal function, had more triple vessel disease, lower left ventricular ejection fraction (LVEF), whereas they were less frequently smokers and presented more often with Non-ST-segment elevation myocardial infarction (Table 1). Treatment characteristics differed substantially between ventilated and non-ventilated patients. Ventilated patients less frequently received manual thrombectomy, and radial access. Additionally, they more frequently received mechanical circulatory support, catecholamines, targeted temperature management, and had longer ICU length of stay and longer time to hemodynamic stabilization (Table 2). 

### 3.2. Clinical Outcomes

The primary endpoint of death at 30 days occurred in 21.0% of non-ventilated patients, in 49.6% of ventilated patients on admission (adjusted OR 5.24, 95% confidence interval [CI] CI 2.74–10.04, *p* = 0.007, compared to patients without MV) and in 61.5% of patients ventilated within 1 day after admission (adjusted OR 7.18, 95%CI 3.81–13.53, *p* < 0.001, compared to patients without MV) (Figure 2, Table 3). After 365 days of follow-up, the primary endpoint occurred in 26.9% of non-ventilated patients, in 55.8% of patients ventilated at admission and in 67.3% of patients ventilated within the first day after admission (*p* < 0.001) (see Appendix A).

On multivariate analysis, independent predictors for MV at admission were higher body weight, left bundle branch block, resuscitation within 24 h before randomization, higher heart rate, catecholamine requirement, and absence of ST-segment elevation. Independent predictors for MV initiated within the first day after admission were higher heart rate and catecholamine requirement (see Appendix A).

A total of 366 (53.6%) patients required cardiopulmonary resuscitation within 24 h before randomization, most of them (*n* = 348, 95.1%) received MV either at presentation (92.8%) or within the first day after admission (7.2%). Among the remaining patients who did not undergo resuscitation (*n* = 315), a total of 101 (32.1%) required no ventilation, 85 (26.9%) were ventilated at presentation and 129 (41.0%) received MV within the first day or later after admission.

Patients without resuscitation requiring ventilation more frequently had, in general terms, greater BMI, creatinine values and heart rate, more frequently presented with triple vessel disease, were more likely to have previous history of PCI, more commonly had clinical signs of impaired organ perfusion, lower LVEF, whereas they were less frequently smokers. The primary endpoint occurred more often among ventilated patients (see Appendix A). Additionally, manual thrombectomy and femoral access were more commonly used in these patients. Mechanical support, catecholamine use and hypothermia due to in-hospital cardiac arrest that occurred after admission were more frequently used in the MV group as well.

### 3.3. Sensitivity Analyses

Among patients who received respiratory support (*n* = 564) where type of ventilation was available (*n* = 503), 56 (11.1%) received NIV alone, 12 (2.3%) received NIV before IMV, and the majority (*n* = 435, 85.5%) received IMV alone. Baseline characteristics of all groups are shown in Appendix A. The primary endpoint was higher among patients requiring IMV (29.1%) and specially among those receiving NIV before IMV (66.7%).

## 4. Discussion

In this secondary analysis of the CULPRIT-SHOCK trial, we investigated the outcomes associated with respiratory failure in patients with CS and AMI and observed three novel findings. First, over 80% of patients developed respiratory failure, with the majority (60.4%) requiring MV already on admission (intubated out-of-hospital). Second, patients receiving MV had an approximately 60% risk of death within 30 days, including a higher observed risk among patients who received MV after admission. Third, over 10% of patients were initially treated with NIV, which was successful in nearly 80%, as defined by no need for further IMV. However, in patients who failed NIV and required IMV, mortality was significantly higher than in those who did not fail NIV.

Since the SHOCK (Should we Emergently Revascularize Occluded Coronaries for Cardiogenic Shock) trial nearly 20 years ago showed the superiority of mechanical revascularization over medical therapy, no interventions, including mechanical circulatory support, have proven to improve outcomes in patients with CS [4,15,16,17,18]. Indeed, there is still an unacceptable high mortality rate, especially among high-risk patients [19,20,21]. One possible reason for the lack of benefit from many of these interventions is the heterogeneity and spectrum of severity in CS [17]. As part of a recent classification schema by the Society for Cardiovascular Angiography and Interventions (SCAI) to improve risk stratification and guide treatment, the need for respiratory support is considered as a marker of severity in CS (Stages C to E) [17,22]. Similarly, in the present study, we found that for patients who required respiratory support, independently of the ventilator strategy (NIV and/or IMV) and timing, respiratory support identified a higher-risk population.

We found that patients ventilated on admission had a lower risk of death compared to those requiring ventilation later, which persisted after the exclusion of patients presenting with cardiac arrest. Similarly, mechanical circulatory support in those undergoing MV within the first day was also more frequent, suggesting greater shock severity in this group. While the etiology for this difference is likely multifactorial, early MV may offer several therapeutic benefits. In addition to improving oxygenation, ventilation, and work of breathing, with the consequent attenuation of the hypoxia-ischemia-hypoperfusion cycle, the use of MV may also provide beneficial effects on cardiovascular hemodynamics. Through an increase in intrathoracic pressure, positive pressure ventilation may lower left ventricular afterload and decrease preload in patients with left ventricular dysfunction [1,23,24], particularly in patients with elevated filling pressures in whom cardiac output can be augmented [25,26]. However, in patients with right ventricular failure, positive pressure ventilation can increase right ventricular afterload and decrease preload, thereby worsening hemodynamics and decreasing cardiac output [1]. These differential effects of positive pressure ventilation highlight the importance of a proper understanding of the heterogeneous spectrum of shock, the identification of right ventricular involvement, as well as the specific cardiopulmonary interactions of specific patient profiles. While our results are hypothesis generating, further research is needed to assess the ideal timing of MV and associations with clinical outcomes for patients with CS, especially given that our results did not allow us to explore associations between MV and outcomes according to the presence of predominantly right or left ventricular shock.

Very little is known regarding the use of NIV in patients with CS [8]. Hongisto et al. categorized ventilatory support strategies in the prospective, multicenter CardShock registry [27]. Among the 219 CS patients, they reported that 63% received IMV, 12% received NIV, and 26% did not need any ventilator support. A total of eight patients failed NIV and required IMV but were split amongst the two ventilator groups (*n* = 4 in both). At 90 days, mortality was 27% and 50% for the NIV and IMV groups, respectively. However, after propensity matching for demographics, medical history, and acuity, the 90-day mortality was similar with both modalities. Of note, the sample size in that registry was too small to assess patients who failed NIV and required IMV [27].

In our larger cohort, we observed that among 11% of patients initially treated with NIV, 78% did not require escalation to IMV. Among patients not escalated to IMV, the 30-day mortality was substantially lower (29%) than in patients who required IMV (64%). Patients in whom NIV failed and required IMV, had the highest mortality (over 83%). While this was a much smaller cohort, our findings are consistent with findings from other critical care populations which reported worse outcomes for patients in whom intubation was delayed, was initially not indicated or in whom NIV failed [28]. In a randomized controlled trial, for patients who developed respiratory distress within 48 h after extubation and were randomized to NIV vs standard oxygen therapy, NIV actually had lower survival rates than those randomized to standard oxygen therapy. They found that those randomized to NIV had a significantly longer interval from respiratory failure to eventual reintubation [29]. The mechanisms behind these findings are likely multifactorial (e.g., cardiac ischemia, complications of emergency intubation, respiratory muscle fatigue, etc.), but highlight the importance of heightened vigilance needed for this critically ill group, and the potential of a false sense of safety that may take place when using NIV, and that could potentially delay transition to IMV and therefore lead to adverse outcomes. Further research to identify the optimal timing of intubation is warranted.

## 5. Limitations

Our study should be viewed in light of several limitations, including the inherent limitations of a post-hoc analysis and the potential for imbalanced variables and unmeasured confounders between comparison groups. First, we were unable to identify the exact timing of respiratory support implementation other than on admission or within the first day after admission. Therefore, we are unable to classify more specific timing such as before, during, or after revascularization or perform an analysis of outcomes based on time delays for implementation of MV or transition from NIV to IMV. Second, data about specific MV parameters, settings or respiratory mechanics such as tidal volume, positive-end expiratory pressure, peak airway pressures, plateau pressures or MV modes are not available in the present database, limiting the ability to draw conclusions about the association of MV settings on outcomes. Third, our study includes only patients with AMI and multivessel disease and may not be generalizable to other etiologies of CS or with AMI and single vessel disease. Fourth, in some patients it was not possible to obtain an informed consent and they had to be withdrawn for the study.

## 6. Conclusions

In the CUPLRIT-SHOCK trial cohort, we observed that patients with AMI complicated by CS who received MV had a significantly higher risk of mortality compared to patients without MV. NIV was utilized in approximately 10% of patients, associated with favorable outcome if successful, but was associated with a mortality over 80% for those who failed NIV and required intubation. Additional research is necessary to identify the optimal timing and ideal respiratory support modality for patients with CS with AMI.

## Figures and Tables

**Figure 1 jcm-09-00860-f001:**
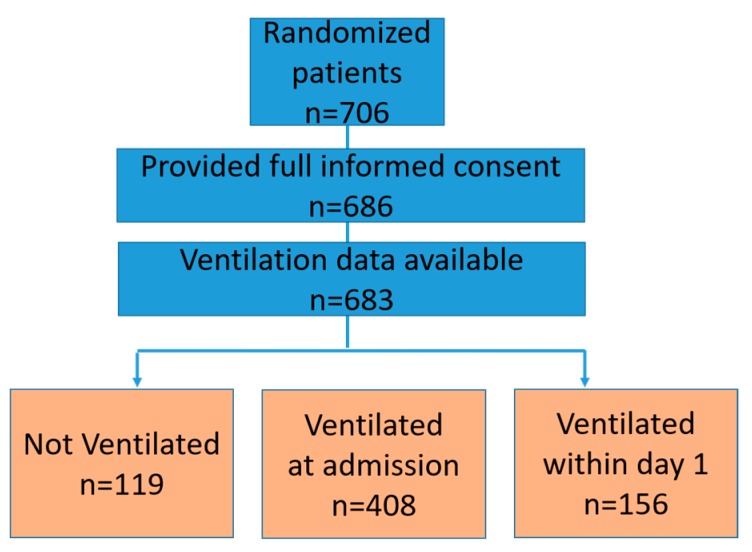
Study flow. Flowchart displaying patients randomized in the study and included in the current analysis.

**Figure 2 jcm-09-00860-f002:**
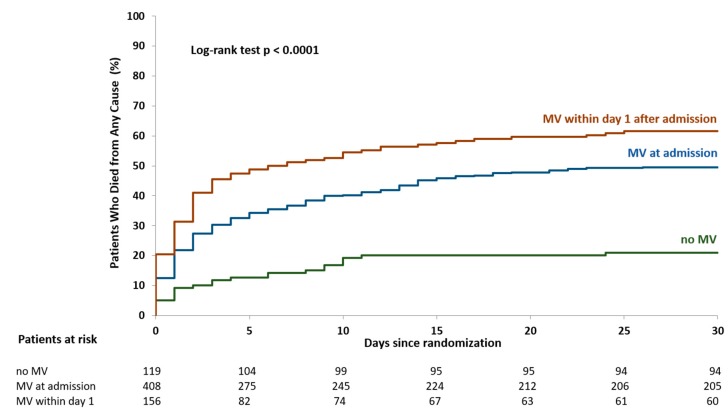
Time to primary endpoint in 30 days. Kaplan‒Meier estimates time-to-event curves for death from any cause in 30 days. MV = mechanical ventilation.

**Table 1 jcm-09-00860-t001:** Baseline characteristics.

	No Ventilation	Ventilation at Admission	Ventilation within Day 1 after Admission	
	*n* = 119	*n* = 408	*n* = 156	*p*-Value
Age‒years				
Median (IQR)	70 (60,78)	68 (59, 77)	73 (63, 80)	< 0.001
BMI kg/m^2^				
Median (IQR)	25.7 (23.4, 28.4)	27.1 (24.7, 29.4)	26.6 (24.5, 30.1)	0.008
Cardiovascular risk factors *n*/*N* (%)				
Current smoking	43/118 (36.4)	96/391 (24.6)	34/148 (23.0)	0.02
Hypertension	69/119 (58.0)	240/401 (59.9)	95/152 (62.5)	0.74
Dyslipidemia	44/119 (37.0)	134/398 (33.7)	48/152 (31.6)	0.65
Diabetes mellitus	38/119 (31.9)	126/398 (31.7)	54/153 (35.3)	0.71
Family history of CAD	19/118 (16.1)	43/385 (11.2)	17/144 (11.8)	0.35
Previous myocardial infarction *n*/*N* (%)	24/118 (20.3)	65/400 (16.3)	24/153 (15.7)	0.53
Previous stroke *n*/*N* (%)	9/119 (7.6)	23/401 (5.7)	17/154 (1.9)	0.12
Known peripheral artery disease *n*/*N* (%)	15/119 (12.6)	40/402 (10.0)	25/154 (16.2)	0.12
Known renal insufficiency (GFR < 30 mL/min) *n*/*N* (%)	6/119 (5.0)	27/402 (10.0)	13/152 (8.6)	0.52
Chronic dialysis *n*/*N* (%)	1/119 (0.8)	2/402 (0.5)	2/154 (1.3)	0.61
Previous PCI *n*/*N* (%)	25/118 (21.2)	81/400 (20.3)	20/153 (13.1)	0.12
Previous CABG no (%) *n*/*N* (%)	4/119 (3.4)	25/402 (6.2)	3/154 (1.9)	0.08
Signs of impaired organ perfusion *n*/*N* (%)				
Altered mental status	54/119 (45.4)	327/406 (80.5)	79/155 (51.0)	<0.001
Cold, clammy skin and limbs	67/119 (56.3)	296/399 (74.2)	105/153 (68.6)	<0.001
Oliguria (≤ 30 mL/h)	12/117 (10.3)	121/394 (30.7)	40/147 (27.2)	<0.001
Arterial lactate > 2.0 mmol/L	49/117 (41.9)	279/396 (70.5)	111/149 (74.5)	<0.001
Resuscitation before randomization *n*/N (%)	18/119 (15.1)	323/408 (79.2)	111/149 (74.5)	<0.001
ST-segment elevation *n*/*N* (%)	85/117 (72.6)	223/392 (56.9)	105/153 (68.6)	0.002
anterior *n*/*N* (%)	40/84 (47.6)	121/221 (54.8)	59/104 (56.7)	0.42
non-anterior *n*/*N* (%)	44/84 (52.4)	100/221 (45.2)	45/104 (43.3)	0.42
ST-segment depression *n*/*N* (%)	45/117 (38.5)	189/392 (48.2)	72/153 (47.1)	0.17
LBBB *n*/*N* (%)	11/117 (9.4)	68/393 (17.3)	20/153 (13.1)	0.08
Mean arterial pressure‒mmHg				
Median (IQR)	78 (63, 93)	75 (63, 91)	75 (63, 95)	0.89
Heart rate-beats/min				
Median (IQR)	79 (61, 101)	91 (78, 108)	94 (72, 109)	< 0.001
Creatinine (mmol/l)				
Median (IQR)	8.8 (7.2, 13.4)	13.1 (9.3, 17.6)	12.3 (9.8, 17.4)	< 0.001
N°of affected vessels *n*/*N* (%)				0.03
1 *n*/*N* (%)	2/119 (1.7)	3/408 (0.7)	0/155 (0.0)	
2 *n*/*N* (%)	54/119 (45.4)	137/408 (33.6)	54/155 (34.8)	
3 *n*/*N* (%)	63/119 (52.9)	268/408 (65.7)	101/155 (65.2)	
Artery with culprit lesion *n*/*N* (%)				0.003
Left anterior descending	45/119 (37.8)	176/408 (43.1)	66/155 (28.4)	
Left circumflex	21/119 (17.6)	98/408 (24.0)	27/155 (17.4)	
Right coronary	46/119 (38.7)	100/408 (24.5)	44/155 (28.4)	
Left main	7/119 (5.9)	27/408 (6.6)	18/155 (11.6)	
Bypass	0/119 (0.0)	7/408 (1.7)	0/155 (0.0)	
Left ventricular ejection fraction‒%				
Median (IQR)	38 (30, 48)	33 (25, 40)	30 (20, 38)	0.01

IQR = interquartile range; BMI = body mass index; CAD = coronary artery disease; GFR = glomerular filtration rate; PCI = percutaneous coronary intervention; CABG = coronary artery bypass graft; LBBB = left bundle branch block.

**Table 2 jcm-09-00860-t002:** Procedural characteristics.

	No Ventilation	Ventilation at Admission	Ventilation within Day 1 after Admission	
	*n* = 119	*n* = 408	*n* = 156	*p*-Value
Arterial access no/total *n*/*N* (%)				
Femoral	85/119 (71.4)	352/408 (86.3)	124/155 (80.0)	<0.001
Radial	34/119 (28.6)	60/408 (14.7)	32/155 (20.6)	0.002
Brachial	0/119 (0.0)	2/408 (0.5)	1/155 (0.6)	0.71
Stent in culprit lesion *n*/*N* (%)				
Any	115/119 (96.6)	391/408 (95.8)	142/155 (91.6)	0.08
Bare metal	6/115 (5.2)	21/391 (5.4)	10/142 (7.0)	0.74
Drug eluting	105/115 (91.3)	373/391 (95.4)	133/142 (93.7)	0.23
Direct Stenting *n*/*N* (%)	27/119 (22.7)	78/408 (19.1)	28/155 (18.1)	0.60
Aspiration thrombectomy before stenting *n*/*N* (%)	27/119 (22.7)	43/408 (10.5)	28/155 (18.1)	0.76
TIMI grade for blood flow *n*/*N* (%)				
Before PCI *n*/*N* (%)				<0.001
0	77/116 (66.4)	191/404 (47.3)	96/153 (62.7)	
I	10/116 (8.6)	56/404 (13.9)	16/153 (10.5)	
II	15/116 (12.9)	67/404 (16.6)	24/153 (15.7)	
III	14/116 (12.1)	90/404 (22.3)	17/153 (11.1)	
After PCI *n*/*N* (%)				0.07
0	1/118 (0.8)	19/405 (4.7)	7/154 (4.5)	
I	1/118 (0.8)	10/405 (2.5)	9/154 (5.8)	
II	10/118 (8.5)	25/405 (6.2)	13/154 (8.4)	
III	106/118 (89.8)	351/405 (86.7)	125/154 (81.2)	
Mechanical support *n*/*N* (%)	25/119 (21.0)	107/408 (26.2)	61/156 (39.1)	0.002
IABP	12/25 (48.0)	19/107 (17.8)	19/61 (31.1)	0.004
Impella 2.5	3/25 (12.0)	21/107 (19.6)	10/61 (16.4)	0.64
Impella CP	5/25 (20.0)	30/107 (28.0)	13/61 (21.3)	0.52
TandemHeart	0/25 (0.0)	2/107 (1.9)	0/61 (0.0)	0.44
ECMO	1/25 (4.0)	25/107 (23.4)	19/61 (31.1)	0.026
Other	4/25 (16.0)	11/107 (10.3)	5/61 (8.2)	0.56
Mild induced hypothermia *n*/*N* (%)	1/119 (0.8)	205/406 (50.5)	23/156 (14.7)	< 0.001
Procedural success (TIMI 3 flow or successful complete revascularization) *n*/*N* (%)	105/108 (97.2)	351/377 (93.1)	126/137 (92.0)	0.21
Duration of mechanical ventilation‒days				
Median (IQR)	*n* = 0	3 (1, 8)	2 (1, 6)	< 0.001
Duration of ICU treatment‒days				
Median (IQR)	3 (2, 6)	6 (2, 12)	4 (1, 11)	< 0.001
Catecholamine requirement *n*/*N* (%)	75/119 (63.0)	391/406 (96.3)	146/156 (93.6)	< 0.001
Duration of catecholamine days				
Median (IQR)	1 (1, 3)	2 (1, 5)	2 (1, 5)	< 0.001
Days to hemodynamic stabilization‒days				
Median (IQR)	1 (1, 4)	3 (1, 6)	2 (1, 8)	< 0.001

TIMI=thrombosis in myocardial infarction; PCI = percutaneous coronary intervention; IABP = intra-aortic balloon pump; ECMO = ExtraCorporeal Membrane Oxygenation; IQR = interquartile range;

**Table 3 jcm-09-00860-t003:** Clinical outcomes at 30 days.

	No Ventilation	Ventilation at Admission	Ventilation within Day 1 after Admission	
	*n* = 119	*n* = 408	*n* = 156	*p*-Value
**Primary endpoint *n*/*N* (%)**				
Death	25/119 (21.0)	202/407 (49.6)	96/156 (61.5)	<0.001
**Secondary endpoints *n*/*N* (%)**				
Renal replacement therapy	9/119 (7.6)	63/407 (15.5)	24/156 (15.4)	0.08
Myocardial Infarction	0/119 (0.0)	7/407 (1.7)	0/156 (0.0)	0.09
Rehospitalization	1/119 (0.8)	1/407 (0.2)	0/156 (0.0)	0.43
Repeat revascularization	22/119 (18.5)	46/407 (11.3)	18/156 (11.5)	0.10
**Safety endpoints *n*/*N* (%)**				
Bleeding event	20/119 (16.8)	94/407 (23.1)	34/156 (21.8)	0.10
Stroke	2/119 (1.7)	12/407 (2.9)	9/156 (5.8)	0.13

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
