# Peer review of "Outcomes Associated with Respiratory Failure for Patients with Cardiogenic Shock and Acute Myocardial Infarction: A Substudy of the CULPRIT-SHOCK Trial"

_jcm, 2020, doi:10.3390/jcm9030860_

Round 1

Reviewer 1 Report

Abstract.
47-48. Currently reframe the sentence. " Little is known about clinical outcome of patient with acute myocardial infarction complicated by cardiogenic shock" . Arthur should justify this sentence.

Line 49-50 . Needs revision.

Line 52. Changed" we explored clinical outcome within 30 days" To 30 days. Our pruritus 30-day mortality/30-day outcome.

Line 56-58 once again needs revision. The sentence is not connected and brings ambulatory to the readers

Background.
69-71
Other should once again specify why patients with acute myocardial infarction and cardiogenic shock were mechanically ventilated. They should also specify whether patient had concomitant pulmonary edema or any other pulmonary disease complicating the acute myocardial infarction.
-What was the main need for mechanical ventilation?

Outcomes.
The primary endpoint should be either all cause mortality or severe renal failure requiring renal respiratory therapy.. As renal failure kidney itself can cause death daughter should try to differentiate these things.

112. What does mechanical traction present on admission ?

Statistical analysis
-Can be returned to bring more clarity to avoid confusion

Results
140 " among the 683 patient's included from the culprit shock trial, 408 presented with mechanical ventilation and admission. After she was specified when they were intubated and why they were intubated.

The authors should also specify patients with mechanical ventilation and without making ventilation including respiratory parameters, arterial blood, chest x-ray/CAT scan, trial of noninvasive mechanical ventilation, who is extensive pulmonary morbid morbidity like COPD.

Clinical outcome
I recommend using MOTILITY as primary endpoint.
-Authors should also specify why renal replacement therapy at 30 days she was used as a part of primary endpoint and not other pulmonary or cardiac related endpoints.

It is interesting, though her respiratory failure is only been used 4 times in the entire manuscript and the aim of the study was to look at the outcome of patients with mechanical ventilation in patient with cardiogenic shock.

Discussion.
Line 205 ? What does mechanical ventilation present on admission remains. Per the patient transfer from the other hospital the patient was intubated due to a cardiac arrest outside the hospital

Author Response

Reviewer 1

Abstract.
47-48. Currently reframe the sentence. " Little is known about clinical outcome of patient with acute myocardial infarction complicated by cardiogenic shock" . Arthur should justify this sentence.

As suggested by reviewer 1, we now reformulated the sentence: Little is known about clinical outcomes of patients with acute myocardial infraction (AMI) complicated by cardiogenic shock (CS) requiring mechanical ventilation (MV).” Abstract, Page 1, Line 48.

Line 49-50 . Needs revision.

As suggested by reviewer 1, we now reformulated the sentence: “The aim of this study was to identify the characteristics, risk factors, and outcomes associated with the supply of MV in this specific high-risk population.” Abstract, Page 2, Line 49-50.            

Line 52. Changed" we explored clinical outcome within 30 days" To 30 days. Our pruritus 30-day mortality/30-day outcome.

As suggested by reviewer 1, we now reformulated the sentence: “We explored 30 days clinical outcomes in patients”. Abstract, Page 2, Line 52.

Line 56-58 once again needs revision. The sentence is not connected and brings ambulatory to the readers.

As suggested by reviewer 1, we now reformulated the sentence: “The primary endpoint of all-cause death occurred in 21% of patients without MV, in 50% of patients with MV at admission and 62% of patients with MV initiated within the first day after admission.” Abstract, Page 2, Line 56-58.

Background.
69-71
Other should once again specify why patients with acute myocardial infarction and cardiogenic shock were mechanically ventilated. They should also specify whether patient had concomitant pulmonary edema or any other pulmonary disease complicating the acute myocardial infarction. -What was the main need for mechanical ventilation?

As suggested by reviewer 1, we now clarify in more detail that respiratory failure was the need for mechanical ventilation: Patients with cardiogenic shock (CS) due to acute myocardial infarction (AMI) experience respiratory failure requiring mechanical ventilation (MV) in 80% of the cases.” Background, Page 2, Line 69-71.

Outcomes.
The primary endpoint should be either all cause mortality or severe renal failure requiring renal respiratory therapy. As renal failure kidney itself can cause death daughter should try to differentiate these things.

As suggested by reviewer 1 we now established 30-day mortality as the primary endpoint in this subanalysis, leaving renal failure requiring renal replacement therapy as a secondary endpoint. The “Methods” section now reads as follows:                                                                                                                                             “The primary endpoint of the current subanalysis was death from any cause within 30 days after randomization”. Methods, Page 3, Line 198-109.

  1. What does mechanical traction present on admission ?            

We are not entirely sure what mechanical traction means. As defined in the “Methods” section, patients were stratified in 3 groups according to the use of mechanical ventilation: 1) no MV (no requiring MV at all), 2) MV present on admission and 3) MV initiated within the first day after admission (or the following days).                                                                             

Statistical analysis: Can be returned to bring more clarity to avoid confusion.

We thank reviewer 1 for his comment about the statistical analysis. Unfortunately, we are not sure about what is meant with “bring more clarity”. Besides, we now rephrased the definition of primary outcome in the statistical analysis section.                                                                                                                            “The primary endpoint of death from any cause within 30 days after randomization is displayed by Kaplan-Meier-curves.” Methods, Page 3, Line 120-121.

Results
140 " among the 683 patient's included from the culprit shock trial, 408 presented with mechanical ventilation and admission. After she was specified when they were intubated and why they were intubated.

As suggested by reviewer 1, we now added that respiratory failure was the reason for the use of mechanical intubation: or later after admission due to respiratory failure.” Results, Page 3, Line 139-140

The authors should also specify patients with mechanical ventilation and without making ventilation including respiratory parameters, arterial blood, chest x-ray/CAT scan, trial of noninvasive mechanical ventilation, who is extensive pulmonary morbid morbidity like COPD.

We thank reviewer 1 for his suggestion. Unfortunately, we do not have data regarding respiratory parameters, imaging or prior respiratory morbidities. We included this limitation in our “Discussion” section as follows:                                                                                                                            “Second, data about specific MV parameters, settings or respiratory mechanics such as tidal volume, positive end expiratory pressure, peak airway pressures, plateau pressures or MV modes are not available in the present database, limiting the ability to draw conclusions about the association of MV settings on outcomes.” Discussion, Page 10, Line 264.

Clinical outcome

I recommend using MOTILITY as primary endpoint.
-Authors should also specify why renal replacement therapy at 30 days she was used as a part of primary endpoint and not other pulmonary or cardiac related endpoints.

As suggested by reviewer 1 we now established 30 days mortality as the primary endpoint in this subanalysis, leaving renal failure requiring renal replacement therapy as a secondary endpoint. The “Methods” section now reads as follows in the revised version:                                                                                                                                             “The primary endpoint of the current subanalysis was death from any cause within 30 days after randomization”. Methods, Page 3, Line 198-109.

It is interesting, though her respiratory failure is only been used 4 times in the entire manuscript and the aim of the study was to look at the outcome of patients with mechanical ventilation in patient with cardiogenic shock.

We are not entirely sure what the reviewer means. However, as suggested by the reviewer , we now used the term “respiratory failure” more frequently. Patients with cardiogenic shock (CS) due to acute myocardial infarction (AMI) experience respiratory failure requiring mechanical ventilation (MV) in 80% of the cases.” Background, Page 2, Line 69-71.

Discussion.
Line 205 ? What does mechanical ventilation present on admission remains. Per the patient transfer from the other hospital the patient was intubated due to a cardiac arrest outside the hospital

As suggested by reviewer 1, we now clarify in more detail that patients presenting already on mechanical ventilation were intubated out of hospital and not transferred from other centers.                                                                                                                                                     requiring MV already on admission (intubated out-of-hospital).” Discussion, Page 9, Line 202-203.

Reviewer 2 Report

I thank the authors for the privilege to review the paper titled: ` Outcomes Associated with Respiratory Failure for Patients with Cardiogenic Shock and Acute Myocardial Infarction: A Substudy of the CULPRIT- SHOCK Trial.´ There is no doubt that the CULPRIT-SHOCK trial was an excellent study with a high numbers of patient included and very important findings. I also think the in terms of methods this subanalysis of the study cohort is proper and correct. Unfortunately I do not see the impact of this study on our field. The finding that mechanical ventilation is associated with worse outcome after myocardial infarction is not novel and these patients represent the highest risk patients within the CS population. I personal do not see the impact of further research to identify the optimal timing for MV in this high-risk patients because if there are so sick that they cannot sufficiently breath, they need ventilation.

Thank you again for giving me the opportunity to comment on the paper,

Sincerely,

Author Response

We kindly appreciate the comments of Reviewer 2.

Reviewer 3 Report

In this secondary analysis of the CULPRIT-SHOCK trial, Authors investigated the outcomes associated with respiratory failure in patients with CS and AMI and observed 3 novel findings. First, over 80% of patients developed respiratory failure, with the majority requiring MV already on admission. Second, patients receiving MV had an approximately 60% risk of death within 30 days. Lastly, over 10% of patients were initially treated with NIV, which was successful in nearly 80%. Requiring MV in patients with CS complicating AMI is common and independently associated with mortality after adjusting for covariates. Patients with delayed MV initiation appear to be at higher risk of adverse outcomes.  The paper is well written and the authors to be congratulated on their efforts. Despite the extensive analysis; this findings to be handled carefully and not generalized in all CS setting.

Author Response

Reviewer 3

In this secondary analysis of the CULPRIT-SHOCK trial, Authors investigated the outcomes associated with respiratory failure in patients with CS and AMI and observed 3 novel findings. First, over 80% of patients developed respiratory failure, with the majority requiring MV already on admission. Second, patients receiving MV had an approximately 60% risk of death within 30 days. Lastly, over 10% of patients were initially treated with NIV, which was successful in nearly 80%. Requiring MV in patients with CS complicating AMI is common and independently associated with mortality after adjusting for covariates. Patients with delayed MV initiation appear to be at higher risk of adverse outcomes.  The paper is well written and the authors to be congratulated on their efforts. Despite the extensive analysis; this findings to be handled carefully and not generalized in all CS setting.

We kindly appreciate the comments of reviewer 3. As suggested, we included in our “Limitations” section that results should not be generalized in all CS settings.

“Third, our study includes only patients with AMI and multivessel disease and may not be generalizable to other etiologies of CS or with AMI and single vessel disease.” Discussion, Page 11, Line 272-273.